# Application of Machine Learning Models in Systemic Lupus Erythematosus

**DOI:** 10.3390/ijms24054514

**Published:** 2023-02-24

**Authors:** Fulvia Ceccarelli, Francesco Natalucci, Licia Picciariello, Claudia Ciancarella, Giulio Dolcini, Angelica Gattamelata, Cristiano Alessandri, Fabrizio Conti

**Affiliations:** Lupus Clinic, Rheumatology, Dipartimento di Scienze Cliniche Internistiche Anestesiologiche e Cardiovascolari, Sapienza Università di Roma, Viale del Policlinico 155, 00161 Rome, Italy

**Keywords:** Systemic Lupus Erythematosus, artificial intelligence, machine learning models

## Abstract

Systemic Lupus Erythematosus (SLE) is a systemic autoimmune disease and is extremely heterogeneous in terms of immunological features and clinical manifestations. This complexity could result in a delay in the diagnosis and treatment introduction, with impacts on long-term outcomes. In this view, the application of innovative tools, such as machine learning models (MLMs), could be useful. Thus, the purpose of the present review is to provide the reader with information about the possible application of artificial intelligence in SLE patients from a medical perspective. To summarize, several studies have applied MLMs in large cohorts in different disease-related fields. In particular, the majority of studies focused on diagnosis and pathogenesis, disease-related manifestations, in particular Lupus Nephritis, outcomes and treatment. Nonetheless, some studies focused on peculiar features, such as pregnancy and quality of life. The review of published data demonstrated the proposal of several models with good performance, suggesting the possible application of MLMs in the SLE scenario.

## 1. Introduction

Systemic Lupus Erythematosus (SLE) is a systemic autoimmune disease, extremely heterogeneous in terms of immunological features and clinical manifestations (Figure 1A). Thus, this condition could potentially involve any organ and system, leading to different severity degrees and outcomes. Traditionally, it is possible to distinguish more severe disease, including renal and neurological manifestations, from mild/moderate disease, characterized by other manifestations such as skin and joint involvement [1]. This clinical complexity could result in a diagnostic delay, especially evident when the disease begins with rarer manifestations. Data from the literature reported an interval between the appearance of first symptom and diagnosis of about 70 months. Of note, it must be underlined that diagnostic delay, even when it amounts to only 6 months, could lead to severe organ involvement, high flare rates and chronic damage development. This derives certainly from the later introduction of appropriate treatment, together with prolonged treatment of glucocorticoids, widely recognized as the most relevant risk factors for chronic damage progression [2,3].

In this view, the purpose of the classification criteria is to facilitate and to anticipate SLE diagnosis and to allow the identification of homogeneous populations for clinical studies. The classification criteria proposed until now have been summarized in Figure 1B [4,5,6]. The latest EULAR/ACR criteria published in 2019 introduced important innovations. First of all, the presence of an entry criterion, represented by the ANA positivity, is necessary to apply these criteria, underlining the autoimmune pathogenesis of SLE. In addition, a weighted score system has been introduced, with different scores for different clinical and laboratory features. Accordingly, only patients reaching a score higher than 10 could be classified as affected by SLE. The application of these criteria leads to a sensitivity and specificity of 98.0% and 96.4%, respectively in the derivation cohort, and of 96.1% and 93.4%, respectively in the validation cohort [6].

From a pathogenic point of view, a multifactorial etiology has been widely demonstrated for SLE. More than one hundred genetic variants have been associated with disease susceptibility and phenotype. Then, the interplay between genetic background and different environmental factors leads to the activation of an aberrant autoimmune response with the production of several autoantibodies [1,7]. The production of autoantibodies has been described several years before the appearance of clinical manifestations, suggesting a stage of subclinical autoimmunity preceding the disease development [8,9].

SLE is traditionally characterized by a relapsing-remitting course, with the occurrence of disease flare. This evolution could result in the development of irreversible chronic damage, determined by disease activity itself and by the adverse events of treatment, in particular glucocorticoids [10,11]. The application of more appropriate therapeutic approaches, in particular the so-called treat-to-target, could significantly impact the course of the disease. In fact, a better control of disease activity, by reaching remission or a low disease activity state, could determine the reduction of chronic damage progression, with improvement in long-term outcome and survival [12]. In 2019, the latest recommendations have been published, based on a comprehensive management of SLE patients. In fact, it is not only the need to treat the disease itself that has been underlined, but also comorbidities, and the need to educate patients to an appropriate lifestyle, with emphasis on sun protection, vaccination, exercise, and smoking cessation [13]. Concerning the pharmacological approach, the recommendations distinguished mild, moderate and severe manifestations to prescribe more appropriate treatment. Of note, the possibility to use a biological treatment, in particular belimumab, was introduced for the first time in the routine care of SLE patients [13]. In the view of disease complexity, several unmet needs are still present for the diagnosis and the management of SLE patients, suggesting the application of innovative tools, such as machine learning models (MLMs). Thus, the purpose of the present review is to provide the reader with information about the possible application of artificial intelligence (AI) in patients with SLE from a medical perspective.

### Methods

A literature search was done in PubMed, accessed via the National Library of Medicine PubMed interface (http://www.ncbi.nlm.nih.gov/pubmed, accessed on 1 December 2022). Firstly, PubMed was searched using the term “systemic lupus erythematosus” OR “lupus” in combination with (AND) “machine learning models”. Secondly, the same PubMed search was combined with other terms, such as “artificial intelligence” OR “classification” OR “clustering” OR “regression”.

## 2. Machine Learning Models: General Concepts

In the last years, AI has generated increasing interest in the field of medical conditions, including rheumatic diseases. In particular MLMs, a subcategory of AI, have been widely applied for different purposes, such as diagnosis, identification of disease phenotypes, prognosis and precision medicine [14]. 

Differently from the statistical method, MLMs extract knowledge from input data. Indeed, if the statistical models aim at explaining specified or hypothesis-driven relationships, MLMs work to search underlying data connections and make decisions according to the newly discovered associations. Thus, MLMs extrapolate relationships unidentifiable with other statistical techniques that are more suitable to generate new hypotheses [15].

The ideal application of MLMs involves the use of so-called big data, deriving from electronic health records, imaging tools, genetics, and transcriptomic procedures. This could be very interesting in the evaluation of complex chronic conditions, such as rheumatic diseases, characterized by great heterogeneity in clinical and laboratory features, by overtime evolution and by the contribution of multiple factors in disease susceptibility and course [15]. Thus, MLMs could help in predicting disease outcome e treatment response, a challenge in diseases such as SLE, characterized by alternating clinical course and various severity degrees requiring different treatments.

The aim of MLMs is the generation of a predictive model, potentially relevant in routine care for the following outcomes: classification, regression or clustering [16,17]. As reported in Figure 2A, different types of data could be used as input to create the MLMs. In particular, it is possible to use clinical/demographic information, laboratory data, results from patient-reported outcomes, data from tissues analysis or imaging tools, response to different treatments, and information about disease activity course or chronic damage development [14,16,17].

The use of medical data in MLMs frequently requires a process of adaptation, in particular they should be translated into a numerical format that could be processed by the AI. Furthermore, it should also be considered the possibility of missing data. Sometimes, a scaling process should be applied to transform existing features into a smaller set of variables [15,18]. Moreover, MLMs perform better when the number of input variables is optimized. Indeed, features selection is a dimensionality technique of reduction that is applied to identify the most appropriate variables to use as input into MLMs algorithm, as all measured variables might not provide information that is necessary for outcome prediction. The features selection could be made by using different modalities, including filter, wrapper and embedded methods [19].

In detail, different types of MLMs are available: to summarize, supervised and unsupervised algorithms can be differentiated according to the labeling of used variables. Thus, a supervised model is constructed to predict known values, whereas an unsupervised model works to predict unknown variables [20,21]. Figure 2B summarizes the different MLMs that could be applied in the supervised and unsupervised modalities [15,20,21]. Of note, the performance of supervised models could be improved by using two independent datasets: the training and the validation dataset. Furthermore, the performance could be assessed by applying different metrics, such as accuracy (the ratio of correct predictions to total predictions), sensitivity (the true positive rate) and specificity (the true negative rate). If the classification problem is binary, these values are often represented by using receiver operating characteristic (ROC) curves. Thus, the area under the ROC curve (AUC) represents the probability that the model can distinguish correct and incorrect outcomes. An AUC value of 1.0 indicates a perfect model performance, whereas a score of 0.5 indicates that the model’s performance is comparable to random chance [22]. For regression analysis, other parameters could be used to assess the model performance: in particular, mean squared error, root mean squared error and the coefficient of determination [23].

Given the availability of different algorithms, it is essential to select the most appropriate MLMs according to the goal (classification, regression, clustering or reduction of dimensionality). Furthermore, the MLMs selection should be based on the available input data, and the comparison between multiple algorithms is recommended in order to identify the model with the greatest performance [24,25].

## 3. Machine Learning Models in SLE Cohorts

In the last years several studies have applied MLMs in SLE cohorts in different disease-related fields. In particular, the majority of studies focused on diagnosis and pathogenesis, disease-related manifestations, outcomes and treatment.

### 3.1. Diagnosis

Table 1 summarizes data about the studies applying MLMs for diagnostic purposes [26,27,28,29,30,31,32,33,34,35,36,37,38,39,40,41]. Overall, it is possible to identify three fields of application according to the data input considered in the studies. First of all, moving from the role exerted by the genetic background in disease development, more recent studies applied MLMs in this context [27,28,29,30]. Therefore, MLMs could be applied to select candidate genes able to identify SLE patients, suggesting the possibility to use these inputs as diagnostic biomarkers. Furthermore, other laboratory features have been considered as input for AI models, such as proteomic data deriving from serum, plasma or peripheral blood mononuclear cells (PBMCs) of SLE patients [26,41]. Already in 2009 Huang and colleagues proposed a Decision Trees model, including a panel of four proteins that resulted able to recognize SLE patients [41]. More recently, Li and colleagues, by using Random Forest model, demonstrated a good performance for a six-protein combination model (SLE versus healthy controls, AUC = 0.7; SLE versus rheumatoid arthritis, AUC = 0.815). The AUC increased up to 0.990 when considering the ability of a nine-protein combination in discriminating SLE patients with disease flare from patients with stable disease [26].

In 2021, Matthiensen and colleagues for the first time applied MLMs to assess the diagnostic role of plasma lipidome, showing good sensitivity and specificity in distinguishing SLE from patients with cardiovascular disease and ischemic stroke [32].

In the remaining studies, the ability of MLMs for a diagnostic purpose has been tested by using electronic health data (EHD) or clinical/laboratory disease-related features, frequently as defined by the available classification criteria. Overall, the use of EHD as input demonstrated a good performance of MLMs in identifying SLE patients in terms of AUC values (up to 0.97) [38,39].

The study published by Adamichou and colleagues in 2020 aimed at assessing the accuracy of 2019 EULAR/ACR criteria in SLE diagnosis by using a LASSO-LR model (ref). The inclusion as input of all the features included in the three classification criteria sets (ACR 1997, SLICC 2012, ACR/EULAR 2019) allowed to observe an accuracy for the most recent criteria of 94.8% in identifying SLE patients. In detail, a higher sensitivity was demonstrated for subjects with an early disease, for patients with Lupus Nephritis (LN) and neuropsychiatric SLE (NPSLE), and for patients treated by immunosuppressant drugs or biological agents. Furthermore, the authors were able to develop a predictive score (the so-called SLERPI score): for a score higher than seven, an accuracy of 94.2% was observed [33]. Our group employed different MLMs—in particular the ReliefF algorithm, Logistic Regression, nonlinear Support Vector Machines, and Decision Trees models—to identify the stronger predictors for SLE diagnosis. By enrolling SLE patients and control subjects with miscellaneous rheumatic diseases, relevant to the differential diagnosis, we obtained a good model’s performance, already when only the three highest scoring features were considered (AUC = 0.94). Furthermore, anti-dsDNA positivity, low C3/C4 serum levels and malar/maculopapular rash resulted in the strongest predictor features for classifying a patient as having SLE [34].

Moreover, the application of cluster analysis could be used to identify subsets of patients by integrating clinical features, immunological profiles and molecular pathways. In this context, the study conducted by Guthridge and colleagues in 2020 used different parameters as input, by combining data from plasma, serum and RNA evaluation with clinical and immunological features. Indeed, the application of a cluster analysis allowed to identify different disease clusters in terms of molecular profile, such as expression of interferon, and disease activity, as assessed by SLEDAI-2k [36]. Similarly, Diaz-Gallo and colleagues in 2022 applied an unsupervised cluster analysis by identifying four SLE subgroups, different in terms of the autoantibody profile, HLA-DRB1 alleles, immunological and clinical features [42].

The same model allowed to differentiate SLE patients according to lymphocyte subsets. Indeed, the study conducted by Lu and colleagues identified four clusters (B high, CD4 high, CD8 high and NK high). These clusters differed in terms of clinical manifestations: in fact, the incidence of arthritis was significantly higher in B high cluster, while nephritis was more frequent in CD8 high and NK high clusters. Finally, CD4 high cluster showed SLEDAI-2k values significantly lower compared with the remaining three clusters [43]. In this view, cluster analysis could also differentiate SLE patients according to cytokine profile: as demonstrated by Reynold and colleagues, it is possible to identify three distinct groups of patients, characterized by higher levels of interferon-alpha and B lymphocyte stimulator (group 1), increased CXCL10 and CXCL13 (group 2) or low levels of cytokines (group 3). Furthermore, group 2 had significantly lower serum complement and higher anti-dsDNA antibodies with increased prevalence of arthritis [44].

### 3.2. Disease Features

The majority of the available studies focused on the application of MLMs on SLE cohorts with renal involvement, representing one of the most fearful disease-related manifestations, with possible progress into end-stage renal disease in 20% of patients and then requiring more aggressive treatment [45]. Table 2 summarizes data about these studies [46,47,48,49,50,51,52], the first of which was published in 2011 and focused on the probability of 3-year allograft survival after renal transplantation [46]. By considering different input data, such as previous and current treatments and data about transplantation and comorbidities, the authors applied different models, obtaining a good performance in terms of AUC (up to 0.74 when considering the logistic regression model) [46]. Only one recent study included, as input, gene expression datasets, downloaded from the GEO database. The application of LASSO and SVM-FRE models suggested the possible role as diagnostic biomarkers for the following genetic variants: C1QA (AUC = 0.741), C1QB (AUC = 0.758), MX1 (AUC = 0.865), RORC (AUC = 0.911), CD177 (AUC = 0.855), DEFA4 (AUC = 0.843), HERC5 (AUC = 0.880) [48]. In the remaining studies on LN, the models used simultaneously clinical and demographic data, serum and urinary biomarkers, and histological features for diagnostic and predictive purposes [47,49,50,51,52]. Two studies published in 2022 demonstrated a good performance of MLMs in discriminating different histological classes. Indeed, Wang and colleagues proposed a model able to distinguish between ISN/RPS pure class V and classes III ± V or IV ± V, while Yang and colleagues observed good accuracy for mask R-CNN and LSTM models on recognizing different glomerular diseases based on slide images (AUC = 0.947) [48,52]. Furthermore, MLMs resulted able to predict a one-year response to treatment, the complete remission, or the risk of flare at 5 years follow-up [47,49,51].

Neurological involvement represents another complex SLE manifestation, with heterogeneous phenotype and lack of specific biomarkers. Thus, the differential diagnosis between SLE-related neurological symptoms and other confounder disorders is not always easy [53]. In this view, MLMs could facilitate clinicians in discriminating the real NPSLE from other pathological conditions. The main aspects of the studies published so far were summarized in Table 2 [54,55,56,57]. In detail, two studies applied MLMs in the analysis of the role of imaging techniques for diagnostic purposes. Thus, cluster analysis resulted able to discriminate five subsets of magnetic resonance characterized by the predominant involvement of different cerebral areas in terms of white matter hyperintensities distribution [55]. Moreover, the application of proton magnetic resonance spectroscopy was evaluated by SVM with feature selection. The authors proposed a diagnostic model with 94.9% of accuracy, which was able to identify patients with early NPSLE [56]. The study conducted by Barraclough and colleagues in 2022 focused on patients with cognitive impairment: the application of the network fusion model is able to discriminate patients with different performances in cognitive functions [57]. Gu and colleagues proposed a model by integrating the presence of anxiety and T-cells subsets evaluated by flow cytometry: the XGBoost model allowed to identify a significant difference in terms of T-cell subsets in patients with or without anxiety (AUC= 0.922) [54].

Two studies conducted by our research group focused on the application of MLMs in SLE-related joint involvement, one of the most frequent manifestation, potentially involving up to 90% of patients [58]. The first study published in 2018 applied logistic regression with the Forward Wrapper method in a cohort of patients with joint involvement evaluated from a laboratory and ultrasonographic point of view. We obtained a model with a good performance in identifying SLE patients with erosive arthritis (AUC = 0.806). Furthermore, at the feature selection, anti-carbamylated proteins antibodies (anti-CarP) resulted the most relevant factors for the presence of erosive arthritis [59]. Thus, an unsupervised hierarchical cluster analysis was applied to identify the aggregation of patients with and without erosive arthritis into different subgroups sharing common characteristics in terms of clinical and laboratory phenotypes. Our results demonstrated the identification of four main clusters: in particular, erosive arthritis was located in a cluster including renal and neuropsychiatric involvement, serositis, positivity for anti-CarP, anti-citrullinated protein antibodies, anti-Sm, anti-RNP, detectable levels of Dkk1 [60]. This could suggest the presence of a more aggressive disease phenotype, sharing a common pathogenic background [61].

**Table 2 ijms-24-04514-t002:** Data about studies applying Machine Learning Models in different disease-related features, in particular Lupus Nephritis, Neuropsychiatric SLE, joint involvement.

Study	Disease Feature	MLM	Input Data	Results
Tang, 2011 [46]	Lupus Nephritis	Classification trees Logistic Regression Artificial Neural Network	Demographic, clinical, laboratory data; treatment, data about transplantation, comorbidity	Model to predict the probability of 3-year allograft survival after renal transplantation. LR, AUC = 0.74 Classification trees, AUC = 0.70 95% CI: 0.67–0.72) ANN, AUC = 0.71
Chen, 2021 [47]	Lupus Nephritis	XGBoost SR-SPM	Clinical, laboratory and histological data	Development of a model to evaluate the risk of renal flare 5 years after remission. Good performance (XGBoost, C-index = 0.819) (SR-SPM, C-index = 0.746)
Wang, 2022 [48]	Lupus Nephritis	LASSO Support Vector Machine	LN gene expression datasets downloaded from the GEO database	Possible role as diagnostic biomarkers for C1QA (AUC = 0.741), C1QB (AUC = 0.758), MX1 (AUC = 0.865), RORC (AUC = 0.911), CD177 (AUC = 0.855), DEFA4 (AUC = 0.843), HERC5 (AUC = 0.880)
Stojanowski, 2022 [49]	Lupus Nephritis	Multi-layer perceptron	Demographic and laboratory features	Development of a predictive models for complete remission, (accuracy = 91.67%, AUC 0.9375)
Wang, 2022 [50]	Lupus Nephritis	HMFO Support Vector Machine	Demographic and laboratory features	Development of a model distinguishing between ISN/RPS pure class V and classes III ± V or IV ± V
Ayoub, 2022 [51]	Lupus Nephritis	Logistic Regression Random Forest Support Vector Machine Artificial Neural Network	Clinical data, urine biomarkers	Development of a model to predict 1-year treatment response (AUC = 0.7)
Yang, 2022 [52]	Lupus Nephritis	Mask R-CNN LSTM	Human kidney biopsy samples	Good accuracies (up to 0.940) on recognizing different glomerular diseases based on H&E whole slide images (AUC = 0.947)
Gu, 2021 [54]	NPSLE	LASSO Random Forest XGBoost	Clinical data, flow cytometry data on T-cell subsets, Self-Rating Anxiety/Depression Scale and Beck Depression Inventory	Identification of difference in T-cell subsets in SLE patients with or without anxiety Best performer XGBoost (AUC = 0.922)
Rumetshofer, 2022 [55]	NPSLE	Cluster analysis	White matter hyperintensities on MRI	Identification of five distinct clusters with predominant involvement of different areas.
Tan, 2022 [56]	NPSLE	Support Vector Machine Feature selection	Proton magnetic resonance spectroscopy	Diagnostic model with 94.9% accuracy, 91.3% sensitivity, 100% specificity and 0.87 cross-validation score
Barraclough, 2022 [57]	NPSLE	Network fusion	Cognitive assessment using the ACR Neuropsychological Battery (ACR-NB)	Identification of two subtypes with different performance in cognitive function (*p* < 0.03)
Ceccarelli, 2018 [59]	Joint involvement	Logistic Regression Forward Wrapper method Feature selection	Clinical and laboratory data, Ultrasound assessment	Good performance to identify patients with erosive arthritis (AUC = 0.806).
Ceccarelli, 2022 [60]	Joint involvement	Cluster analysis	Clinical and laboratory data, Ultrasound assessment	Identification of four clusters. Erosive arthritis was located in a cluster including renal and NPSLE.

LR: Logistic Regression; AUC: Area Under Curve; ANN: Artificial Neural Network; ISN/RPS: International Society of Nephrology (ISN)/Renal Pathology Society (RPS).

Furthermore, MLMs have been also applied in the field of SLE comorbidity. In detail, Liu and colleagues in 2022 used AI to identify potential biomarkers for SLE patients with atherosclerosis (AS). By applying LASSO, SVM-RFE, and RF models, the authors identified five hub genes (specifically, SPI1, MMP9, C1QA, CX3CR1, and MNDA) with a high predictive performance in distinguishing subjects with and without AS (AUC ranging from 0.900 to 0.981) [62]. Wang and colleagues aimed at identifying the shared genes between SLE and metabolic syndrome (MetS): RF and LASSO algorithms were used to screen shared hub genes, and a diagnostic model was created by applying XG-Boost. Finally, the authors identified shared hub genes and constructed an effective diagnostic model in SLE and MetS. In detail, TNFSF13B and OAS1 had a positive correlation with cholesterol and xenobiotic metabolism. Both biomarkers and metabolic pathways were potentially linked to monocytes, providing novel insights into the disease pathogenesis [63].

### 3.3. Disease Activity and Damage

The main outcome in the management of SLE patients is certainly the control of disease activity in order to prevent chronic damage development. The longitudinal assessment of disease activity allowed to identify different patterns: the so-called relapsing-remitting pattern has been prevalently associated with damage progression, due to the need to use glucocorticoids to treat disease relapse [12]. Several efforts have been made to develop tools able to properly measure disease activity, but the failure of the majority of randomized controlled trials enrolling SLE patients suggests that this field represents still an unmet need [64]. In this view, MLMs could play a potential role.

In 2018 the study published by Toro-Dominguez aimed at stratifying SLE patients in terms of disease activity according to gene expression. The application of cluster analysis allowed to identify three different clusters in pediatric and adult patients; furthermore, in one cluster the authors observed a significant correlation between neutrophils percentage and a lower disease activity, evaluated by SLEDAI [65]. Furthermore, by using a real-world dataset, MLMs resulted able to discriminate SLE patients with different SLEDAI values, when using a cut-off equal to five [66].

Other studies proposed the integration of clinical data with gene expression, also providing suggestions for pathogenic mechanisms implicated in determining disease activity. In this field, Kegerreis and colleagues proposed a Random Forest model with an accuracy equal to 83% in discriminating patients with active and inactive disease according to genetic profile [67]. More recently, rule-based machine learning models and rule networks were applied to develop gene networks to separate pediatric SLE patients according to a state of low and high disease activity. The authors proposed a model with a good performance (accuracy 81%) to distinguish different levels of disease activity. Furthermore, the application of unsupervised hierarchical clustering revealed additional subgroups characterized by the association between specific gene pathways and disease activity. In detail, the following genetic variants have been clustered: IFI35 and OTOF; KLRB1 encoding CD161; CKAP4 [68].

Interestingly, cluster analysis was applied to identify an association between risk flare and peripheral immunophenotypes, as assessed by flow cytometry. Thus, the so-called memory B-cells cluster showed a lower risk to develop disease flares compared with the non-memory B-cells group, including naïve B- and T-cells [69].

In 2017 our research group applied recurrent neural networks to predict chronic damage development, assessed by the SLICC Damage index (SDI) [70]. Thus, for the Recurrent Neural Network model we selected two groups of patients: patients with SDI = 0 at the baseline, developing damage during the follow-up, and those without damage during the whole follow-up. By using these data inputs, we could create a model with an AUC value equal to 0.77, able to predict damage development. A threshold value of 0.35 (sensitivity = 0.74, specificity = 0.76) seemed able to identify patients at risk to develop damage [71]. More recently, in the study conducted by Ahn and colleagues, cluster analysis allowed to identify three groups of patients according to the damage severity and mortality risk [72]. The relationship between damage clustering and mortality was previously evaluated by Pego-Regoisa et al. in a large Spanish SLE cohort. Overall, the authors identified three clusters according to the severity of damage, two of them showed a significantly higher mortality rate [73].

MLMs were recently used by our group to propose a new outcome in SLE field: the so-called Lupus comprehensive disease control (LupusCDC), including both the achievement of remission and the absence of damage progression [74]. The proposal of LupusCDC originates from the evidence that the control of disease activity is not always sufficient to stop the damage progression, due to the presence of other factors concurring with its development [11,75]. Thus, we applied SVM models and Decision Trees, followed by features ranking with the ReliefF algorithm. Our model, characterized by AUC value equal to 0.703, identified glucocorticoids, renal involvement and the use of immunosuppressant drugs as the most relevant factors concurring to the failure to achieve LupusCDC [74].

### 3.4. Treatment

In the last years the concept of precision medicine has been widely spread in the context of rheumatic conditions, including SLE. The heterogeneity of this disease, possible expression of different underlying pathogenic mechanisms, suggests the need for personalized treatment according to the most relevant manifestation [76]. In this context, MLMs could help the clinician in the treatment choice, by predicting drug response. However, very few data are available on this specific topic. In 2016 Kan and colleagues evaluated a large newly diagnosed SLE cohort by using cluster analysis: 10 treatment clusters were identified and the most common consisted of minimally treated patients (42.8%). In this cluster, hydroxychloroquine monotherapy, glucocorticoid monotherapy, and corticosteroid/hydroxychloroquine combination therapy were received by 34.0%, 11.2%, and 7.8% of patients, respectively [77]. More recently, Carter and colleagues observed that response to RTX in non-European SLE patients was lowest in an interferon-low, neutrophil-high cluster and highest in a cluster with high expression across all signatures (*p* < 0.001) [78]. Wang et al. applied MLMs to predict the effect of sirolimus on disease activity in 103 SLE patients. The so-called Emax model was selected for MLMs, where the evaluation indicator was the change rate of SLEDAI from the baseline value. The authors concluded that in order to achieve a better therapeutic effect (80% Emax, plateau), maintaining a concentration of 8–10 ng/mL sirolimus for at least 6–12 months was necessary [79]. Finally, recently MyPROSLE, an omic-based analytical workflow for measuring the molecular portrait of individual patients to support clinicians in their therapeutic decisions has been proposed. This is a machine learning-based classification model aiming at assessing the association between dysregulation in immunological response, clinical manifestations, prognosis, flare and remission events and response to Tabalumab. The model MyPROSLE allowed to molecularly summarize patients in 206 gene-modules, clustered into nine main lupus signatures. Preliminary results suggest that the dysregulation of certain gene-modules is strongly associated with specific clinical manifestations, the occurrence of relapses or the presence of long-term remission and drug response. Thus, the authors suggest the possible use of this model to predict clinical outcomes, including treatment response [80].

### 3.5. Pregnancy

SLE mostly affects women of childbearing age and as widely demonstrated, it could be associated with unfavorable pregnancy outcomes. Furthermore, fetal complications, in particular, fetal death and neonatal lupus syndrome could develop in SLE. Finally, disease flare during and after pregnancy is a common complication, with a prevalence ranging from 35% to 70% of patients [81]. In the last years, the pre-gestational counseling and the multidisciplinary approach adopted in the daily clinical practice allowed SLE patients to experience even more uncomplicated pregnancies [82]. Nonetheless, it is very important to select factors able to identify patients at risk of maternal–fetal complications.

In this context, the possible role of MLMs have been evaluated by two recent studies. Thus, Deng and colleagues applied Random Forest, support vector machine-recursive feature elimination and least absolute shrinkage with selection operator to identify genetic biomarkers for adverse pregnancy outcomes. The model identified three feature genes, specifically SEZ6, NRAD1, and LPAR4. Among these, SEZ6 showed the highest in-sample predictive performance, with an AUC value equal to 0.753 [83]. Moreover, Fazzari and colleagues confirmed, by using MLMs, the role of antihypertensive medication use, low platelets, SLE disease activity and lupus anticoagulant positivity as risk factors for adverse events during pregnancy. In detail, the authors evaluated a large SLE cohort by applying different models, in particular Logistic regression with stepwise selection, LASSO, Random Forest, neural network, Support Vector Machines, gradient boosting and SuperLearner. The best performance in terms of AUC was observed for LASSO model (AUC = 0.78) [84].

### 3.6. Other Possible Application

The increasing interest in the possible role of MLMs in SLE cohorts was demonstrated by the application of these tools in other disease-related fields.

Jorge and colleagues applied decision tree, Random Forest, naïve Bayes and logistic regression to predict hospitalization in SLE patients. By analyzing 1996 patients, 4.6% of them were hospitalized in the most recent year of follow-up, the authors demonstrated a good performance for Random Forest model (AUC = 0.751) in predicting hospitalization. Furthermore, anti-dsDNA positivity, low C3 levels, blood cell counts, and increased inflammatory biomarkers, as well as age and albumin, represented the most relevant risk factors for hospitalization [85].

Finally, Margiotta and colleagues used MLMs in the evaluation of quality of life in SLE patients by using cluster analysis. This approach allowed to distinguish different patterns of quality of life, characterized by the prominent involvement of mental or physical components, as assessed by Short-Form 36 (SF-36) [86].

Furthermore, the same MLM was able to identify different clusters related to sleep disorders in SLE subjects, by integrating data deriving from the Pittsburgh Sleep Quality Index and those from SF-36 and anxiety scores [87].

## 4. Conclusions

In conclusion, the present review focused on the possible application of MLMs in the SLE scenario. In particular, MLMs have been applied in the field of diagnosis, pathogenic mechanisms, definition of different disease features and courses, and finally treatment response. As demonstrated by our literature revision, several models have been proposed, revealing good performance in terms of accuracy and AUC.

These results suggest several possible future applications for MLMs. Among these, the application of specific models could help the physicians to identify patients at risk to develop more aggressive disease phenotypes, and thus could guide in the choice of a more appropriate treatments. Nonetheless, MLMs could be used to predict different phenomena, including the response to treatment, thus finding a place in the so-called precision medicine.

However, the application of MLMs in a real-life context finds some obstacles and may still be anticipatory. First of all, they require studies for internal and external validation, secondly it is mandatory to test the MLMs reliability and reproducibility. In the SLE scenario, the studies published so far are characterized by some limitations, such as the sample size of the analyzed cohorts and the lack of replication studies. These aspects certainly do not allow the use of these models in a real-life context.

## Figures and Tables

**Figure 1 ijms-24-04514-f001:**
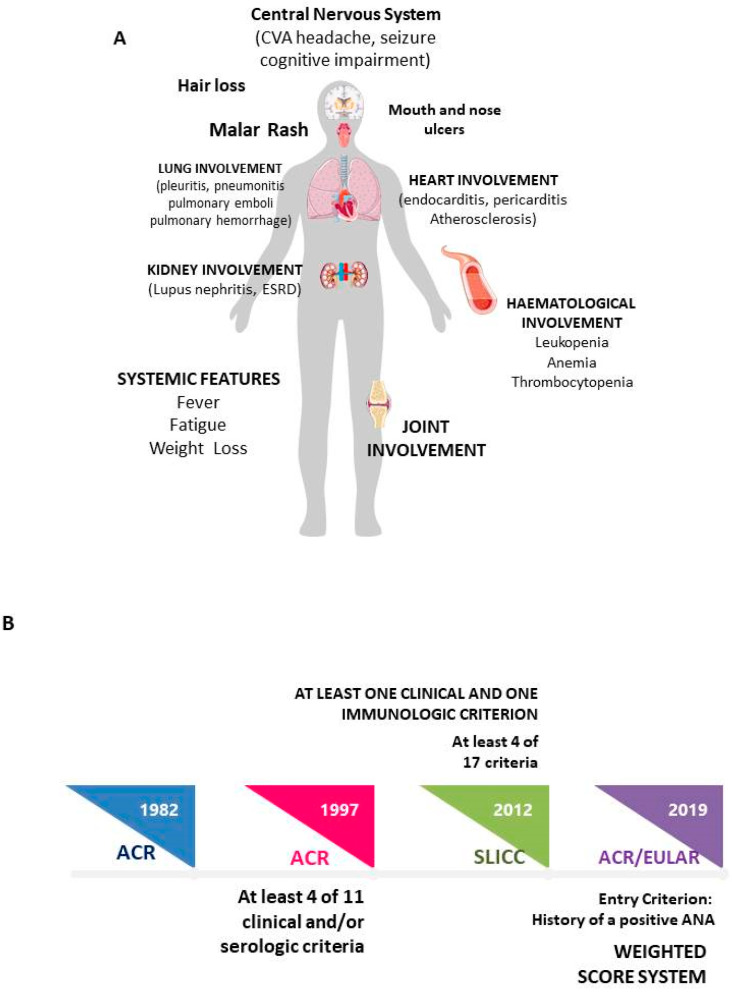
(**A**) Systemic Lupus Erythematosus: main clinical manifestations; (**B**) Classification criteria timeline.

**Figure 2 ijms-24-04514-f002:**
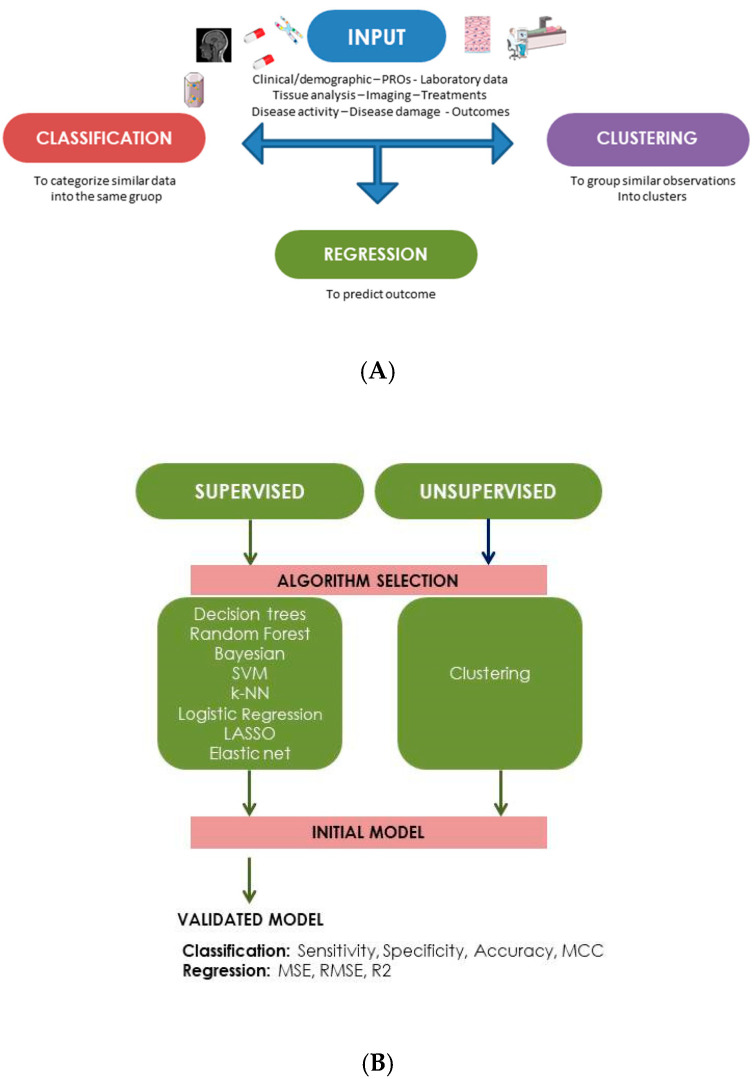
(**A**) Summary of input data used in machine learning models for medical studies, aiming at evaluating the following outcomes: classification, regression or clustering. (**B**) Summary of the different MLMs that could be applied in the supervised and unsupervised modalities.

**Table 1 ijms-24-04514-t001:** Main data about the studies applying Machine Learning Models for diagnostic purposes.

Study	MLM	Input Data	Results
Li, 2022 [26]	Random Forest	PBMC proteomics	Six-protein combination (IFIT3, MX1, TOMM40, STAT1, STAT2, and OAS3) exhibited good performance for SLE disease diagnosis (AUC= 0.723 versus HC; AUC= 0.815 versus RA). Nine-protein combination (PHACTR2, GOT2, L-selectin, CMC4, MAP2K1, CMPK2, ECPAS, SRA1, and STAT2) showed a robust performance in assessing disease exacerbation (AUC = 0.990)
Zhong, 2022 [27]	LASSO Support Vector Machines	Differentially expressed genes (DEGs)	Selection of six candidate diagnostic biomarkers for SLE (ABCB1, EIF2AK2, HERC6, ID3, IFI27, and PLSCR1), with AUC from 0.96 to 0.913
Jiang, 2022 [28]	Logistic regression Random Forest XGBoost Support Vector Machines Artificial Neural Network	Genetic biomarkers from GSE65391 and GSE72509 datasets	IFI44 was determined to be the optimal diagnostic biomarker of SLE
Ma, 2022 [29]	Random Forest Support Vector Machines Artificial Neural Network	Genome-wide association studies	RF model AUC = 0.84 At the optimal cut-off, the RF predictor reached a sensitivity of 84% with a specificity of 68% in SLE classification.
Martorell-Marugán, 2022 [30]	XGBoost model	Biomarkers for gene expression and DNA methylation	The model is able to discriminate SLE from Sjogren Syndrome (gene expression MCC = 0.5791 ± 0.0409; methylation data MCC = 0.5546 ± 0.0484)
Barnado, 2022 [31]	Random Forest XGBoost model	Electonic Health Data	PPV 74–77%
Matthiesen, 2021 [32]	Partial Least Square	Plasma lipidomes	SLE vs CVD (Sensitivity = 0.91, Specificity = 1) IS vs SLE (Sensitivity = 1, Specificity = 0.82)
Adamichou, 2020 [33]	LASSO Logistic Regression	Clinical/laboratory features according classification criteria EULAR/ACR	Accuracy = 94.8% for identifying SLE High sensitivity for early disease (93.8%), LN (97.9%), NPSLE (91.8%), SLE requiring immunosuppressives/biologics (96.4%). Development of a scoring system (>7, 94.2%) accuracy
Ceccarelli, 2021 [34]	ReliefF algorithm, Logistic Regression Support Vector Machines DT models	Clinical/laboratory features according classification criteria EULAR/ACR	At the ReliefF model, anti-dsDNA positivity, low C3/C4 serum levels and malar/maculopapular rash resulted the strongest predictor features. A good model’s performance was obtained already when only the three highest scoring features were considered (AUC = 0.94)
Park, 2020 [35]	Cluster Analysis	Serum cyotkines	Cluster analysis revealed two distinct patient groups characterized by high levels of IL8, MIP1α and MIP1β (group 1) or of IL2, IL6, IL10, IL12, IFNγ and TNF (group 2). Active disease was more common in group 1 (55.7%) than in group 2 (34.8%). More patients in group 2 had renal involvement (42/115, 36.5%) than in group 1 (22/88, 25%).
Guthridge, 2020 [36]	K-means clustering Random Forest	Data from plasma, serum, RNA, clinical, laboratory features	Identification of 7 SLE clusters. Inflammation and interferon modules were elevated in Clusters 1 (moderately) and 4 (strongly), with decreased T-cell modules in Cluster 4. Active clinical features were similar across clusters. Clinical SLEDAI trended highest in Clusters 3 and 4, though Cluster 3 lacked strong interferon and inflammation signatures. Renal activity was more frequent in Cluster 4, and rare in Clusters 2, 5, and 7. Serology findings were lowest in Clusters 2 and 5.
Jorge, 2019 [37]	Logistic Regression	Registration data according classification criteria ACR/SLICC	PPV 90% for definite SLE PPV 92% for definite/probable SLE
Murray, 2019 [38]	Logistic Regression	Electronic health record	AUC=0.97 to automate identification of SLE patients
Turner, 2017 [39]	Artificial Neural Network Random Forest Naïve Bayes model Support Vector Machines Word2Vec	Electronic health record	ICD-9 accuracy 90.00% (AUC = 0.9) Shallow neural network with CUIs accuracy 92.10% (AUC = 0.970) Random forest with BOWs accuracy 95.25% (AUC = 0.994) Random forest with CUIs accuracy 95.00% (AUC = 0.979) Word2Vec inversion accuracy 90.03% (AUC = 0.905)
Dai, 2010 [40]	k-nearest neighbors	Serum peptidome patterns	Blinded verification of the classification model showed 91.7% sensitivity in active SLE, 83.3% sensitivity in stable SLE, and 86.7% specificity in normal controls.
Huang, 2009 [41]	Decision Trees	Serum proteomic	A panel of four potential protein biomarkers could accurately recognize 25 of 32 patients with SLE, 36 of 42 patients with other autoimmune diseases and 36 of 40 healthy people.

SLE: Systemic Lupus Erythematosus; RA: Rheumatoid Arthritis; AUC: Area Under Curve; RF: Random Forest; MCC: Matthews correlation coefficient; PPV: positive predictive value; CVD: cardiovascular disease; IS: ischemic stroke; LN: Lupus Nephritis; NPSLE: neuropsychiatric SLE.

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
