# Peer review of "Application of Machine Learning Models in Systemic Lupus Erythematosus"

_ijms, 2023, doi:10.3390/ijms24054514_

Round 1
Reviewer 1 Report
In general, this article is well-written and easy to follow. It covers Machine Learning applications in Systemic Lupus Erythematosus. The categorisation of Machine Learning applications into subcategories also was good: 1. Diagnostic purposes, and 2. Different disease related features (LN, NPSLE, joint involvement).
Major comments
Figures are good, but can be improved.
A PRISMA pipeline would be suggested so that one easily can follow the review process and article selection: filtering, inclusion/exclusion criteria, etc.
I would also advise to include a RoB assessment.
Could the authors stress more on the future perspective?
Also, could the authors talk more about the reliability and robustness of different ML methods?
Minor comments
In Figure 1A, CNS is misspelled.
Box colours in Figure 2B may be misleading. Hierarchical and k-means clustering, what do they really represent?
Author Response
REVIEWER 1
In general, this article is well-written and easy to follow. It covers Machine Learning applications in Systemic Lupus Erythematosus. The categorisation of Machine Learning applications into subcategories also was good: 1. Diagnostic purposes, and 2. Different disease related features (LN, NPSLE, joint involvement).
Major comments
Figures are good, but can be improved.
Response: Thank you for your response, I modified the figures according with your suggestion in order to improve the usefulness for the readers. I hope that the new version could me more appealing.
A PRISMA pipeline would be suggested so that one easily can follow the review process and article selection: filtering, inclusion/exclusion criteria, etc. I would also advise to include a RoB assessment.
Response: Our purpose was to provide a narrative review that could be of interest for the researchers to clarify potential usefulness of machine learning in the SLE scenario. Thus, we have not conducted a systematic review with meta-analysis and consequently we have not applied advanced statistical analysis, such as RoB assessment. According with your suggestion, we have specified the method for the article selection, now reported in the text, at the end of Introduction section.
Could the authors stress more on the future perspective?
Response: We thanks the referee for this suggestion. Accordingly, we added the following sentence at the end of the Conclusions section: “These results suggests several possible future application for MLMs. Among these, the application of specific models could help the physicians to identify patients at risk to develop more aggressive disease phenotypes, and thus could guide in the choice of a more appropriate treatments. Nonetheless, MLMs could be used to predict different phenomena, including the response to treatment, thus finding a place in the so-called precision medicine”.
Also, could the authors talk more about the reliability and robustness of different ML methods?
Response: We thank the referee for this suggestion. Now, we added a specific sentence about the need for studies to test internal and external validation and for MLMs reliability and reproducibility.
Minor comments
In Figure 1A, CNS is misspelled.
Response: We corrected the typing error.
Box colours in Figure 2B may be misleading. Hierarchical and k-means clustering, what do they really represent?
Response: We changed the figure by modifying the colours. Now we reported one box for clustering analysis.
Reviewer 2 Report
This manuscript described many studies about SLE, which have been conducted through the machine learning method. Although it has valuable aim and can provide a large amount of scientific information, it has several important amendments.
1. The form or content of the manuscript is not well organized. It is not understood what the studies described in the paragraph or section sharing common things. In review papers, similar studies, should be described together, such as dividing them into several categories and producing similar or contradictory results. It needs to consider how to effectively explain many studies that have been conducted with various research materials and methods.
2. In Introduction, it would be better to emphasize how important to diagnose SLE properly without delay and assess disease status in minimizing organ damage and improving the outcomes. That point is the essential aim to summary and review machine learning models about SLE in this manuscript.
3. Methodology including searching strategy, data selection, and extraction including the exclusion criteria should be added.
4. In 3.1 subtitle, “patients clustering” should be removed, because patients clustering methods were used in many studies in other parts.
5. In Conclusion, most of contents are unnecessary and should be mentioned in Introduction (before the review), and it would be better to summarize this review.
6. It would be better to get out the limitation part in Abstract.
7. Figure 1 is not associated with the contents of this manuscript.
8. Line 339-341 should be re-checked because ref 69 is not a study using the machine learning methods.
Author Response
REVIEWER 2
This manuscript described many studies about SLE, which have been conducted through the machine learning method. Although it has valuable aim and can provide a large amount of scientific information, it has several important amendments.
- The form or content of the manuscript is not well organized. It is not understood what the studies described in the paragraph or section sharing common things. In review papers, similar studies, should be described together, such as dividing them into several categories and producing similar or contradictory results. It needs to consider how to effectively explain many studies that have been conducted with various research materials and methods.
Response: We thank the referee for the suggestion. However, in the light of disease heterogeneity we decided to group the studies published so far according to the addressed topic, in order to help the readers to understand the possible application of machine learning models in a such complex disease. Thus, even though the same model have been applied to evaluate different outcomes or disease related phenotypes, we decided to propose a more clinical than methodological point of view, also suitable for non-expert researcher in MLM.
- In Introduction, it would be better to emphasize how important to diagnose SLE properly without delay and assess disease status in minimizing organ damage and improving the outcomes. That point is the essential aim to summary and review machine learning models about SLE in this manuscript.
Response: Thank for this interesting suggestion. Now this aspect has been more ficused in the Introduction section.
- Methodology including searching strategy, data selection, and extraction including the exclusion criteria should be added.
Response: Thank for the suggestion. Accordingly we have specified the method for the article selection, now reported in the text, at the end of Introduction section.
- In 3.1 subtitle, “patients clustering” should be removed, because patients clustering methods were used in many studies in other parts.
Response: We corrected according to the suggestion.
- In Conclusion, most of contents are unnecessary and should be mentioned in Introduction (before the review), and it would be better to summarize this review.
Response: We thank the referee and we modified accordingly the Conclusions section.
- It would be better to get out the limitation part in Abstract.
Response: We thank the referee and we modified accordingly the Abstract section.
- Figure 1 is not associated with the contents of this manuscript.
Response: We decided to insert these figures because we think that this review could be useful also for physician without experience on SLE. Thus, to better clarify the application on MLMs in such complex disease, we inserted these figures.
- Line 339-341 should be re-checked because ref 69 is not a study using the machine learning methods.
Response: We check the reference 69 and we confirmed the use con cluster analysis in this study.